# The Impact of Frailty Components and Preoperative Mechanical Cardiac Support Changes with Time after Heart Transplantation

**DOI:** 10.3390/biomedicines12051114

**Published:** 2024-05-17

**Authors:** Rita Szentgróti, Dmitry Khochanskiy, Balázs Szécsi, Flóra Németh, András Szabó, Kinga Koritsánszky, Alexandra Vereb, Zsuzsanna Cserép, Balázs Sax, Béla Merkely, Andrea Székely

**Affiliations:** 1Doctoral School, Semmelweis University, 1085 Budapest, Hungary; szentgroti.rita@semmelweis.hu (R.S.);; 2Faculty of Medicine, Semmelweis University, 1085 Budapest, Hungary; 3Department of Anesthesiology and Intensive Therapy, Semmelweis University, 1082 Budapest, Hungary; 4Heart and Vascular Centre, Semmelweis University, 1122 Budapest, Hungary; 5Department of Oxiology and Emergency Care, Faculty of Health Sciences, Semmelweis University, 1088 Budapest, Hungary

**Keywords:** heart transplantation, frailty, mechanical cardiac support, frailty screening tool

## Abstract

Background: Frailty has been proven to be associated with mortality after orthotopic heart transplantation (OHT). The aim of our study was to determine the impact of frailty on mortality in the current era using pretransplant mechanical cardiac support (MCS). Methods: We retrospectively calculated the frailty scores of 471 patients undergoing OHT in a single institution between January 2012 and August 2022. The outcome was all-cause mortality. Results: The median survival time was 1987 days (IQR: 1487 days) for all patients. In total, 266 (56.5%) patients were categorized as nonfrail, 179 (38.0%) as prefrail, and 26 (5.5%) as frail. The survival rates were 0.73, 0.54, and 0.28 for nonfrail, prefrail, and frail patients, respectively. The frailty score was associated with mortality [HR: 1.34 (95% CI: 1.22–1.47, *p* < 0.001)]. Among the components of the frailty score, age above 50 years, creatinine ≥ 3.0 mg/dL or prior dialysis, and hospitalization before OHT were independently associated with mortality. Continuous-flow left ventricular assist devices (CF-LVAD) were associated with an increased risk for all-cause mortality [AHR: 1.80 (95% CI: 1.01–3.24, *p* = 0.047)]. Conclusions: The components of the frailty score were not equally associated with mortality. Frailty and pretransplant MCS should be included in the risk estimation.

## 1. Introduction

Orthotopic heart transplantation (OHT) is considered the gold standard therapy for end-stage heart failure when all other pharmacological and nonpharmacological treatment options have been exhausted [1,2]. The assessment of potential OHT recipients is a comprehensive, multistep process. Given an acceptable indication for OHT, compelling evidence demonstrates its ability to enhance survival rates significantly, facilitate a return to everyday activities, improve functional capacity, and thus increase the overall quality of life [2].

According to international guidelines, when establishing the indication for OHT, it is advisable to consider frailty, which has been proven to influence postoperative outcomes [2,3]. Frailty is a complex condition in which the body exhibits increased susceptibility to harmful effects and has difficulty restoring (or is completely unable to restore) homeostasis. However, it is recognized that frailty is not a unidirectional process but a likely reversible condition; with appropriate interventions, it is possible to reduce or even eliminate frailty [4,5,6].

Assessing frailty can be beneficial in evaluating patients awaiting OHT. By identifying and addressing reversible factors, some patient conditions can be optimized before surgery, thereby mitigating the risk of early and late postoperative adverse outcomes. Recently, it has become increasingly accepted that frailty encompasses not only physical but also psychological, psychiatric, and social aspects [7]. Defining and measuring frailty, however, is still a complex task. It is most accurate to envision it as a spectrum without a single well-defined threshold at which an individual can be considered frail [8]. Given this complexity, there is no universally used and accepted method for measuring frailty. Over the years, numerous research groups have developed new measurement tools, questionnaires, and scoring systems to assess frailty [9,10].

In the current era, mechanical cardiac support (MCS) devices have exhibited increasing importance in bridge-to-transplant therapy; approximately 50% of waiting list patients receive one before OHT [2], so it is worth examining whether they have any effect on frailty.

The aim of this study was to evaluate frailty with the OHT frailty screening tool created by Seese et al. [11] and to investigate its effect on postoperative outcomes in our center. Furthermore, we added the different types of MCS devices, which were implanted in the preoperative period, to the original OHT screening tool as further parameters.

## 2. Materials and Methods

The institutional database (Heart and Vascular Centre, Semmelweis University, Hungary) of adolescent and adult patients who had undergone OHT between 1 January 2012 and 11 August 2022 was used for this retrospective analysis. The last follow-up for surviving patients was 1 January 2024. This study was approved by the Regional Ethics Committee (SE RKEB 13/2023).

The dataset contained 471 patients. For those who had two OHTs during one indexed hospitalization, the first transplantation was used. The dependent variable length was measured as days after operation until before death if a patient’s death was observed or after operation until 1 January 2024 if a patient’s death was not observed (right censoring). A total of 149 patients died, which were considered “events”. All other patients left the experiment through right censoring. Appendix A shows the CONSORT statement of this study.

The frailty score was calculated by a 12-point OHT frailty screening tool with the following items: age (≥51 years), body mass index (BMI, <18.5 kg/m^2^ or >31 kg/m^2^), comorbidities in the patient history (diabetes, prior cerebrovascular accident, prior malignancy, recent hospitalized status, recent infection requiring intravenous antibiotics or ventilator dependence), preoperative laboratory values (creatinine level ≥ 3.0 mg/dL or prior dialysis, serum albumin level < 3.5 mg/dL, serum total bilirubin level ≥ 3.0 mg/dL), and functional status based on the Karnofsky performance score (total assistance with activities of daily living—patients who are disabled, severely disabled, very sick or moribund). One point was assigned for each of the above criteria. 

BMI was calculated based on body weight (kg) and body height (cm) with the metric system formula (weight (in kilograms)/height^2^ (in meters)). One point was assigned based on the calculated BMI score if it was <18.5 kg/m^2^ or >31 kg/m^2^ (so an underweight and an overweight group were created).

In our study, one point was assigned for functional status to those who were hospitalized directly before heart transplantation and needed IABP and temporary MCS. The functional status of the nonhospitalized patients was determined by their previous medical records and by the admission records. During admission, the need for assistance, wheelchair use, etc. were explored in detail.

According to the calculated frailty scores, three cohorts were identified: nonfrail (0–2 points), prefrail (3–5 points), and frail (6 or greater) [11]. All the patient data were collected directly from the period of admission for OHT to assess the pretransplant frailty score. Laboratory values were assessed up to 24 h before OHT; any further data were from the latest patient history.

To supplement the original OHT frailty screening tool we recorded the use of MCS devices in the pretransplant period, including short-term MCS (venoarterial extracorporeal membrane oxygenation (ECMO), CentriMag central biventricular assist device (BIVAD)), continuous-flow left ventricular assist device (CF-LVAD, including HeartWare, Heartmate 2, and Heartmate 3), and intra-aortic balloon pump (IABP).

The end point was the length of survival until either death (event) or end of observation (right censoring) at 1 January 2024. As death after OHT should be reported to the central registry of the National Blood Institute, we had information about each transplanted patient.

For the statistical analysis, descriptive statistics, correlation analysis, and models for survival analysis were used. All continuous or ordinal variables are shown as the mean with standard deviation if normally distributed and as the median with interquartile range if not normally distributed or highly skewed. All categorical data are shown as numbers and percentages. We compared continuous variables with Tukey’s test for pairwise mean comparisons and categorical variables with a χ^2^ test. For the association analysis, Cramér’s V and Chi^2^ were used. To estimate the survival functions, the Kaplan–Meier estimator was used, and the survival functions were compared using the log-rank test with Holm–Bonferroni multiple-comparison correction. To evaluate the importance of the variables for survival models, the Cox proportional hazards model was selected with Harrell’s concordance index as a metric for model performance. Individual and combined feature fitting were employed. Additionally, for the Cox survival model, a two-sided *p* value with respect to the null hypothesis that the hazard rates across all groups are equal was used. To measure the importance of features at different time points, time-dependent areas under the ROC and Cox survival models at different follow-up times were used. K-fold cross-validation was employed for combined and individual variable performance of the survival models and time-dependent ROC-AUC.

For statistical analysis, Python 3.10.4 packages, including Pandas 1.5.3, Scikit-Learn 1.1.3, Scikit-Survival 0.19.0, and Lifelines 0.27.4, were used. Figures were generated with MatPlotLib 3.7.1 and Seaborn 0.12.2.

## 3. Results

We calculated the frailty scores of 471 patients who underwent OHT between January 2012 and August 2022 in our center. The mean age of the recipients was 50.79 ± 10.97 years, and 123 patients (26%) were female.

According to the OHT frailty screening tool results, patients were divided into three cohorts: 266 patients (56.5%) were nonfrail, 179 patients (38.0%) were prefrail, and 26 patients (5.5%) were frail.

The most common diagnosis in all frailty categories was ischemic cardiomyopathy (310 patients, 65%). Frail and prefrail patients were older than the nonfrail patients. Frail patients had higher creatinine levels or needed dialysis before OHT and had lower albumin levels. Furthermore, frail patients were more likely to be ventilator dependent, be hospitalized, require total assistance, and need antibiotics in the preoperative period. 

A total of 43 patients (9.3%) received any type of short-term MCS in the pretransplant period. Ten patients had IABP and five patients were on ECMO as the last device therapy before OHT. Of the 28 patients who had BIVAD, 15 received BIVAD upgrade from ECMO, 4 patients had IABP before BIVAD implantation, and in 9 patients BIVAD was the first bridging therapy. The need for short-term MCS in the preoperative period was more frequent in the prefrail and frail patient groups.

In total, 30 patients were on CF-LVAD, among whom 20 patients were hospitalized prior to the transplantation. 

The demographic data, the distribution of the components of the OHT frailty screening tool, and the need for preoperatively implanted MCS devices are shown in Table 1.

The median survival time was 1987 days (IQR: 1487) for all patients. The survival rates were 0.73, 0.54, and 0.28 for nonfrail, prefrail, and frail patients, respectively. The Kaplan–Meier curve for frailty categories is shown in Figure 1. The log-rank value was *p* < 0.001. The starting point of the Kaplan–Meier curve was the day of OHT. 

The frailty score was associated with mortality [hazard ratio (HR) 1.34 (95% CI: 1.22–1.47), *p* < 0.001].

In the multivariable Cox-model, age above 50 years [HR: 1.46 (95% CI: 1.02–2.08, *p* = 0.04)], creatinine ≥ 3.0 mg/dL or dialysis [HR 1.76 (95% CI: 1.24–2.49); *p* = 0.002], hospitalization before OHT [HR: 1.81 (95% CI: 1.22–2.67); *p* = 0.003], and CF-LVAD [HR: 1.81 (95% CI: 1.01–3.24, *p* = 0.047)] were associated with increased risk for mortality. The Cox proportional hazards model for overall mortality only fit with the frailty score with a c-index of 0.64; using components of the frailty score yielded a c-index of 0.67, and adding MCS to this model increased the c-index to 0.68 (Figure 2).

We analyzed each component of the OHT frailty screening tool to determine their individual impact on short- and long-term mortality (results are shown in Appendix A). Age above 50 years was proven to be an independent cause of mortality until the second postoperative year and overall mortality. Recent hospitalization and higher creatinine levels or prior dialysis were significantly associated with postoperative 1-year, 2-year, 5-year, and overall mortality. 

We found that CF-LVAD patients with a recent infection requiring intravenous antibiotics in the immediate pretransplantation period had a high risk of post-transplant mortality in the multivariable model [AHR: 2.59 (95% CI: 0.39–2.79); *p* = 0.01].

The impact of the individual components of the OHT frailty screening tool and MCS had different contributions over time. The plot of cross-validated unadjusted areas under the time-dependent receiver operating characteristic curves for top-performing frailty component predictions is shown in Figure 3.

Multivariable time-dependent Cox hazard models showed that using a continuous age variable in place of a binarized variable slightly increased the performance of the model. Binarized creatinine level or dialysis yielded a better c-index at all times over the continuous variable. The albumin level showed better performance as a continuous variable (Figure 4).

## 4. Discussion

We found that prefrail and frail states were significantly associated with increased mortality after OHT. The predictive value of the frailty score exhibited reduced discriminatory power one year after transplantation. Using the original OHT screening tool by Seese et al. [11] alone had lower discriminative ability. Among the MCS modalities, CF-LVAD was associated with all-cause mortality. In the MCS model, age above 50 years, preoperative creatinine levels ≥ 3.0 mg/dL or prior dialysis, and hospitalization at the time of OHT were independently associated with mortality. During the follow-up period, the impact of the frailty components changed over time.

Comparing our data to Seese et al.’s, not all elements of the OHT frailty screening tool were independently associated with mortality. By itself, the original OHT frailty screening tool had a lower discriminative ability and thus needed to be supplemented in our research, but mortality was equally significantly unfavorable for frail patients in both studies.

An assessment of frailty should be a key element of the preoperative risk stratification in OHT because of its unfavorable effect on mortality and morbidity. In addition, the prevalence of frailty is high in the population of heart failure patients [12,13], so it would be helpful to use it as a general prognostic factor to screen those patients who have a higher risk for worse outcomes. The latest guidelines of the International Society for Heart and Lung Transplantation (ISHLT) recommend the assessment of frailty, but in the absence of a standardized frailty screening tool, the use of the modified Fried frailty criteria, which includes five physical domains (fatigue, gait speed, hand grip strength, unintended weight loss, and physical activity), and the Montreal Cognitive Assessment to detect cognitive function is recommended [2]. Although these two tools together have been proven to be appropriate to assess frailty, their disadvantage is the time required in the busy clinical environment. The modified Fried criteria also include some subjective points, such as the presence of fatigue and current physical activity. In addition, unintentional weight loss cannot be interpretable in patients with end-stage heart failure because of fluid retention, which may mask true weight loss, so it is often replaced by a question about recent loss of appetite, which is also a subjective criterion [14,15,16]. Furthermore, patients with a high-urgency status are unlikely to be able to take part in these investigations, which makes it difficult to assess their frailty states precisely. In our study population, the Seese score based on demographic, disease-related, and laboratory parameters has shown that this approach can be applied in a retrospective manner, but the discriminative value of the individual components varied over time. MacDonald et al. also showed that pretransplant MCS and the frailty status should be measured in parallel and MCS itself is not a promise for the improvement of the frailty state in all cases [16]. Another alternative is the creation and application of a screening tool that is based on easily available data from the patient history and laboratory results. The first frailty screening tool that met these criteria was tailored for OHT recipients [11]. This screening tool offers the advantage of more accurately measuring an acute deterioration in health status. This approach can be applied currently, since 60% of patients who have been recently transplanted were in a high-urgency status, which also means that more MCS devices are being implanted before OHT [17].

Unlike geriatric life expectancies, patients with end-stage heart failure can reclaim their quality of life after transplantation. However, irreversible kidney dysfunction can diminish or further progress in the post-transplantation period, and severe renal dysfunction is associated with significantly lower survival [18]. If inefficient hemodynamics is the main cause of kidney dysfunction, this may improve after OHT, so it is a process that can be reversed to some extent. Patients referred to OHT frequently have comorbidities, such as diabetes mellitus, hypertension, and atherosclerosis, which may worsen their kidney function, and these factors may cause irreversible damage to the kidneys [19]. The role of immunosuppressive medication in worsening kidney disease cannot be ruled out [20]. An older age may also have consequences in rehabilitation, as the incidence of frailty increases with age. Aging, as well as frailty, is associated with several physiological changes that in themselves lead to a reduction in functional reserves and a reduced ability to respond to stressors, such as surgical interventions, which can lead to a higher rate of postoperative morbidity and mortality [21,22]. On the other hand, nutritional status has no impact on medium-term survival, which contradicts previous results in patients after cardiac surgery [23].

Recent changes in the urgency status have changed the general conditions of patients undergoing OHT. While CF-LVAD patients undergoing OHT have favorable and comparable outcomes with the non-MCS recipients, those who had complications associated with the MCS, such as driveline infection, bleeding, or thromboembolic adverse events, will have an increased risk for one-year post-OHT mortality [2].

Based on our results and previous results from the literature, the following question must be answered: what severity of frailty must patients present to undergo heart transplantation? Patients undergoing OHT may have comorbidities in addition to end-stage heart failure; OHT is now available for the elderly as well [3,24], and all these factors are also associated with frailty [25]. Therefore, it is important to screen and detect frailty in this patient population, as frailty is a progressive, nonpermanent condition, and early detection is key to delaying or even reversing the process with a tailored therapeutic intervention called prehabilitation. Prehabilitation includes settlement of nutritional status, improving physical condition, cognitive training, and psychological counseling in the preoperative period [26]. The components of the OHT frailty screening tool tested in this study may also help to eliminate pretransplant risk factors.

The exact limit of reversibility of frailty is not yet known [26]. Therefore, due to the dynamic nature of frailty, it is not possible to determine a clear cut-off for the extent at which OHT should be considered contraindicated. In addition, a recent prospective study has shown that OHT can also lead to the reversal of frailty [27].

## 5. Limitations

First, this study was a retrospective single-center study that included patients with a uniform ethnic and racial background. Only 20 patients from our dataset were classified as frail, which decreased the statistical power of tests performed on this cohort. Retransplantation was performed in only four patients, which also prevented us from evaluating this group. Furthermore, the applied screening tool lacks screening for cognitive impairment, which has been proven to be an important factor in frailty [24]; adding the assessment of cognitive function strengthens the capacity of frailty identification [28]. We also did not gather data about education, employment status, and other sociodemographic characteristics of patients which may have an effect on their frailty status. We cannot rule out the negative influence of the COVID-19 pandemic on mortality, which might be different in patients with different frailty categories. Throughout this period, we observed elevated overall mortality rates [29].

## 6. Conclusions

In conclusion, frailty assessment is an important tool for risk estimation and mortality in the contemporary era. Cardiogenic shock or critical patient conditions can further worsen the risk for mortality after transplantation, but according to our research, severe preoperative conditions and renal dysfunction before OHT have proven to have a negative effect on postoperative survival. The OHT frailty screening tool helps to assess frail patients and may contribute to the reversal of the frail state due to targeted treatment of several morbid conditions. Furthermore, the type of MCS used as part of the bridge-to-transplant therapy should be considered.

## Figures and Tables

**Figure 1 biomedicines-12-01114-f001:**
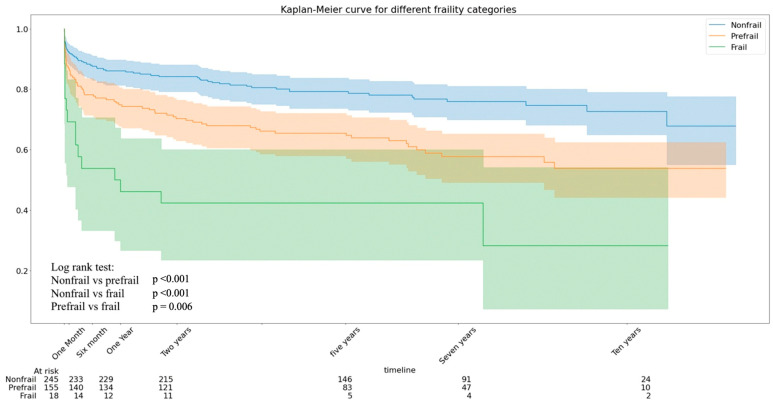
Kaplan–Meier curve for frailty categories.

**Figure 2 biomedicines-12-01114-f002:**
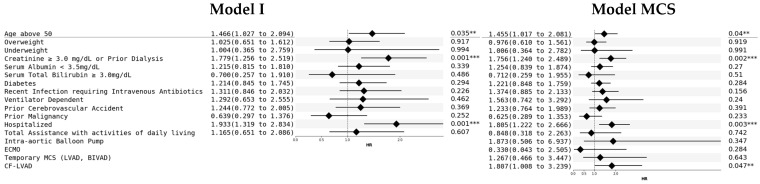
Cox proportional hazards model results for the components of the frailty screening tool for all-cause mortality, with and without the inclusion of mechanical cardiac support. C-index of the original model (Model I): 0.67, c-index of the model with the addition of mechanical cardiac support devices (Model MCS): 0.68. BIVAD: biventricular assist device; CF-LVAD: continuous-flow left ventricular device; ECMO: extracorporeal membrane oxygenation; LVAD: temporary left ventricular assist device; MCS: mechanical cardiac support; Temporary MCS: temporary mechanical cardiac support. The threshold of 0.05 was selected for significance, ** and *** show the level of significance (**: 0.05; ***: 0.01).

**Figure 3 biomedicines-12-01114-f003:**
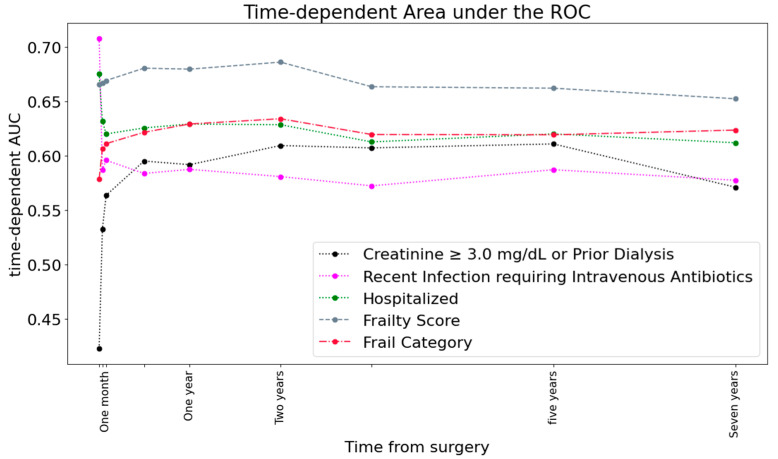
Plot of cross-validated unadjusted areas under the time-dependent receiver operating characteristic curves for top-performing frailty component predictions over mortality outcome.

**Figure 4 biomedicines-12-01114-f004:**
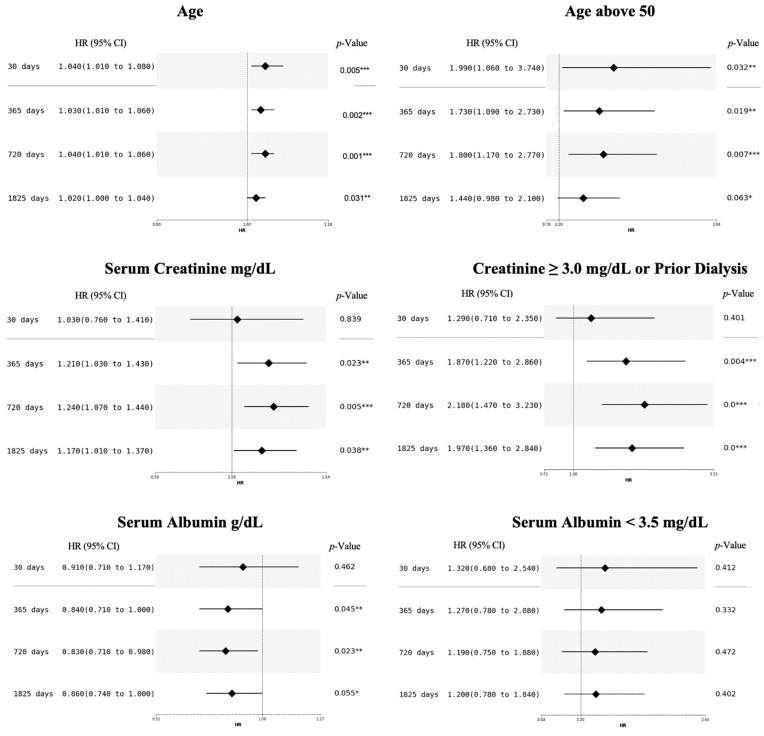
Hazard plots of the multivariable time-dependent Cox hazard model on continuous and binarized variables. The threshold of 0.05 was selected for significance, *, ** and *** show the level of significance (*: 0.1; **: 0.05; ***: 0.01).

**Table 1 biomedicines-12-01114-t001:** Demographic data, distribution of the components of the orthotopic heart transplant frailty screening tool, and need for preoperatively implanted mechanical cardiac support devices.

Variables	All Categories *n* = 471	Nonfrail *n* = 266 (56.5%)	Prefrail *n* = 179 (38.0%)	Frail *n* = 26 (5.5%)	*p*-Values		
					Nonfrail vs. Pre-Frail	Nonfrail vs. Frail	Prefrail vs. Frail
Age (Years)	50.8 ± 11.0	49.3 ± 11.1	52.9 ± 10.3	52.0 ± 11.5	<0.005	0.47	0.91
Sex (Female)	123 (26%)	69 (26%)	49 (27%)	5 (19%)	0.82	0.82	0.82
**Diagnosis**							
Ischemic	310(65%)	188(70%)	103(57%)	19(76%)	1.0	1.0	1.0
Dilated	110(23%)	48(17%)	57(31%)	5(19%)	0.01	0.97	0.29
Restrictive	38(8%)	20(7%)	17(9%)	1(4%)	<0.005	1.0	0.42
Congenital	13(2%)	10(3%)	2(1%)	1(4%)	0.49	1.0	1.0
**Frailty Score Components**							
Age above 50	283 (60%)	131 (49%)	131 (73%)	20 (76%)	<0.005	0.02	0.87
Overweight	74 (15%)	21 (8%)	46 (26%)	7 (26%)	<0.005	0.008	1.0
Underweight	15 (3%)	8 (3%)	5 (3%)	2 (7%)	1.0	0.74	0.74
Creatinine ≥ 3.0 mg/dL or prior dialysis	117 (24%)	34 (13%)	67 (37%)	16 (61%)	<0.005	<0.005	0.03
Albumin < 3.5 mg/dL	93 (19%)	20 (7%)	51 (28%)	22 (84%)	<0.005	<0.005	<0.005
Total bilirubin ≥ 3.0 mg/dL	13 (2%)	2 (1%)	10 (6%)	1 (3%)	0.016	0.95	1.0
Diabetes	141 (29%)	49 (18%)	81 (45%)	11 (42%)	<0.005	0.01	0.94
Recent infection and intravenous antibiotics	88 (18%)	10 (4%)	58 (32%)	20 (76%)	<0.005	<0.005	<0.005
Ventilator dependent	28 (5%)	2 (1%)	12 (7%)	14 (53%)	<0.005	<0.005	<0.005
Prior cerebrovascular accident	60 (12%)	24 (9%)	32 (18%)	4 (15%)	0.02	0.64	1.0
Prior malignancy	27 (5%)	7 (3%)	16 (9%)	4 (15%)	0.008	0.009	0.50
Total assistance	50 (10%)	3 (1%)	32 (18%)	15 (57%)	<0.005	<0.005	<0.005
Hospitalized	183 (38%)	47 (18%)	110 (61%)	26 (100%)	<0.005	<0.005	<0.005
Frailty score	2.48 (1.59)	1.34 (0.70)	3.63 (0.81)	6.19 (0.40)			
**Mechanical Cardiac Support**							
IABP	10 (2%)	1	8 (4%)	1	0.02	0.63	1.0
ECMO	5 (1%)	0 (0%)	3 (2%)	2 (7%)	0.19	<0.005	0.24
BIVAD	28 (5%)	1	16 (9%)	11 (42%)	<0.005	<0.005	<0.005
CF-LVAD ‡	30 (6%)	13 (5%)	14 (8%)	3 (11%)	0.50	0.50	0.79

‡ CF-LVAD Continuous-flow left ventricular assist device: HeartWare, Heartmate 2, Heartmate 3. BIVAD: biventricular assist device; ECMO: extracorporeal membrane oxygenation; IABP: intra-aortic balloon pump.

## Data Availability

All the important and relevant data are included in the Materials and Methods, Results, and Appendix A sections. Please contact the corresponding author if any further data are needed.

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
