# Peer review of "The Impact of Frailty Components and Preoperative Mechanical Cardiac Support Changes with Time after Heart Transplantation"

_biomedicines, 2024, doi:10.3390/biomedicines12051114_

Round 1

Reviewer 1 Report

Comments and Suggestions for Authors

In the article biomedicines-2985799 entitled The impact of frailty components and preoperative mechanical cardiac support changes with time after heart transplantation, Dr. Szentgróti and colleagues studied a single-centered retrospective study of 471 patients who received orthotopic heart transplantation (26% female, mean age of 50.8 years) focusing at their frailty evaluated with the preestablished frailty score (Seese et al. Ann Thorac Surg 2021, PMID: 32622795). They found that the frailty score was significantly associated with postoperative mortality.

Although the topic of this manuscript is interesting, there should be some important points to be reconsidered. Critiques are described below.

1.     Regarding the frailty scoring, it should be explained in more detail about functional status based on the Karnofsky performance score. How one point was assigned for this criteria?

2.     Despite underweight seems reasonable for frail, why overweight (BMI > 31 kg/m2) could also be one of the components for the frailty scoring system? Because the number of the patients with overweight (n = 74, 15%) were larger than the number with underweight (n = 15, 3%), it should be explained in more detail.

3.     Looking at the KM curves (Fig 1), most frail patients reached at endpoint just after the HTx. To ascertain the meanings of the results, it should be clarified how the starting point of the KM curves were defined; on admission, on the HTx operation or the discharge after HTx. Because frail patients received more mechanical support (particularly BIVAD) than other patients, it should be also discussed whether there were significant differences of preoperative physical condition and hospitalization duration in more detail.

4.     Peter S Macdonald et al. had already reported that frailty is significantly associated with mortality for patients with HTx (J Heart Lung Transplant. 2021 PMID: 33279391, as Ref #16). The authors claimed that the frailty scoring tool in the current study was more useful than those used in the previous studies, but the details were not deeply discussed. Comparative discussions should be conducted to determine how the scoring tools used in prior studies were not useful in clinical practice and what the particular advantages of the present scoring system were.

Author Response

Dear Editor, 

We would like to thank you for your generous comments and, as a result, the improvement in the quality of our research article.

Please see the attachment to read our answers.

Yours sincerely, 

Rita Szentgróti

Reviewer 2 Report

Comments and Suggestions for Authors

I would like to thank the authors for permitting me to give an overview of their work, herein I present a few details that would potentially help the overall work.

The write-up contains a few grammatical errors and awkward phrasing. Here are some examples, please revise the full document:

·         "Prefrail states were associated with significantly increased mortality after OHT." - Consider reprasing: "Prefrail and frail states were significantly associated with increased mortality after OHT."

·         "value of the frailty score had less discrimination power one year after transplantation." - Consider reprasing: "The predictive value of the frailty score exhibited reduced discriminatory power one year after transplantation."

·         "screening tool has the advantage of being able to measure acute deterioration in health status more accurately." - Consider reprasing:"This screening tool offers the advantage of more accurately measuring acute deterioration in health status."

·         "During these years, we observed higher overall mortality rates." - Consider reprasing: "Throughout this period, we observed elevated overall mortality rates."

·         "In contrast to geriatric life projections, patients with end-stage heart failure can fully regain their quality of life after transplantation." - Consider reprasing: "Unlike geriatric life expectancies, patients with end-stage heart failure can fully reclaim their quality of life after transplantation."

In addition

would the authors please explain about the BMI is it 2 indices or a range and if so for calculation how were the points calculated for score

Did you measure BNP and troponin levels in relation to heart function?

von Willebrand factor (vWF), and endothelin-1, could be evaluated if possible, to assess vascular health and its relationship with frailty, would be worthwhile considering

Psychosocial needs are crucial for understading patient well-being, while not the focus was there any evaluation.

Comments on the Quality of English Language

For the most part english i feel is ok, just double check a few phrases

Author Response

(The authors gave the same response as above.)

Round 2

Reviewer 2 Report

Comments and Suggestions for Authors

Good efforts in edition